# Stabilization of Natural Pigments in Ethanolic Solutions for Food Applications: The Case Study of *Chlorella vulgaris*

**DOI:** 10.3390/molecules28010408

**Published:** 2023-01-03

**Authors:** Andreia S. Ferreira, Liliana Pereira, Feliciana Canfora, Tiago H. Silva, Manuel A. Coimbra, Cláudia Nunes

**Affiliations:** 1LAQV-REQUIMTE, Department of Chemistry, University of Aveiro, 3810-193 Aveiro, Portugal; 23B’s Research Group, I3Bs—Research Institute on Biomaterials, Biodegradables and Biomimetics, Headquarters of the European Institute of Excellence on Tissue Engineering and Regenerative Medicine, University of Minho, AvePark-Parque de Ciência e Tecnologia, 4805-017 Guimarães, Portugal; 3ICVS/3B’s-PT Government Associate Laboratory, 4710-057 Guimarães, Portugal; 4CICECO–Aveiro Institute of Materials, Department of Materials and Ceramic Engineering, University of Aveiro, 3810-193 Aveiro, Portugal

**Keywords:** *Chlorella vulgaris*, ethanolic extract, pigments, chlorophyll, color stability, food ingredients

## Abstract

*Chlorella vulgaris* is a green microalga with a high chlorophyll content, representing a valuable source of green pigments for food applications. As the application of whole biomass can promote an unpleasant fish-like flavor, the use of chlorophyll extract can overcome this drawback. However, chlorophylls tend to easily degrade when out of the chloroplasts, decreasing their potential as a food ingredient. Thus, to study the suitable conditions for isolated chlorophylls preservation, in this work, the influence of temperature (4 to 60 °C), light (dark or 24 h photoperiod), alkaline conditions (with or without aqueous NaOH addition), and modified atmosphere (air or argon atmosphere) on the stability of the color in ethanolic solutions obtained from *C. vulgaris* were studied. The loss of green color with temperature followed the first-order kinetics, with an activation energy of 74 kJ/mol. Below 28 °C and dark conditions were suitable to preserve isolated chlorophylls. The addition of NaOH and an inert argon-rich atmosphere did not exhibit a statistically positive effect on color preservation. In the case study, cooked cold rice was colored to be used in sushi. The color remained stable for up to 3 days at 4 °C. Therefore, this work showed that *C. vulgaris* chlorophylls could be preserved in ethanolic solutions at room or lower temperatures when protected from light, allowing them to obtain a suitable natural food ingredient to color foodstuffs.

## 1. Introduction

Chlorophylls are green pigments that have been used in the food industry as a natural food ingredient in processed foods. Additionally, due to their strong green pigmentation and consumers’ demand for natural and sustainable foods, following a clean-label market trend, chlorophylls are gaining major importance as food coloring ingredients [1]. Indeed, color plays a dominant role in the appearance and acceptance of food products. Moreover, the change of color in natural and processed food products could be perceived by consumers as a loss of quality, acting as an indicator of proper storage conditions [2]. The green microalga *Chlorella* is a great source of chlorophylls, namely chlorophylls *a* (Chl *a*) and *b* (Chl *b*), being one of the highest chlorophyll contents found in nature, reaching up to 45 mg per g of dry weight [3]. Thus, *Chlorella* species, such as *Chlorella vulgaris*, can be added to processed foods as a green coloring agent. However, the addition of green microalgae to food formulations could potentially develop food products with a slight fish flavor that can be negatively perceived by consumers [4]. In this sense, the liposoluble chlorophylls can be extracted from thylakoid membranes of chloroplasts of *C. vulgaris* with organic solvents [5,6,7]. Ethanol is the most suitable to ultimately use the green extract for food applications. Moreover, the isolation of these chlorophylls leaves a high protein and carbohydrate residue, which can also be potentiated as food ingredients, following a biorefinery-like approach, representing an increase in the revenue stream of *C. vulgaris* [8]. 

Chlorophylls are sensitive compounds since they can degrade easily by various mechanisms when exposed to organic solvents, pH variations, heat, oxygen, or light. Acidification and/or thermal processing result in a perceivable discoloration of chlorophylls from bright green to an olive-green or olive-yellow color, known as pheophytinization. During this reaction, occurs the substitution of the magnesium ion in the porphyrin ring by two hydrogen ions, transforming the natural chlorophylls to their corresponding pheophytins [2,9]. The saponification of the phytyl side chain of chlorophylls can also occur to yield chlorophyllides or pheophorbides [10] (Figure 1). Other chemical degradation routes are oxidation or photo-oxidation if light is implicated, resulting in bleached degradation products [2].

Consequently, efforts in preserving the green chlorophylls mostly in foods [11,12,13,14,15,16] have been carried out through pH control [11,12,16], enzymatic treatments [14], temperature control [11,12,13,16], and the addition of metal ions Cu^2+^ and Zn^2+^ [11,15]. Studies have been focused on the evaluation of the green color stability in aqueous or in organic solvent solutions [17,18]. 

This study aims to obtain a proper green extract to incorporate into food products. Thus, *C. vulgaris* pigments were extracted with ethanol, a food-grade solvent. The influence of combined storage conditions, namely temperature, light, atmosphere, and alkaline environment, on the stability of ethanol solutions of *C. vulgaris* chlorophylls/color was studied. These aim to set the most appropriate storage conditions for *C. vulgaris* ethanolic extract to stabilize the color to further allow their application as a suitable green coloring ingredient without affecting the organoleptic characteristics of the food products.

## 2. Results and Discussion

### 2.1. Evaluation of the Efficiency of Different Solvents to Extract C. vulgaris Pigments

The liposoluble chlorophylls are usually extracted with organic solvents such as chloroform, methanol, ethanol, and acetone or their mixtures [5]. Although ethanol is not the most efficient solvent to extract chlorophylls, this was the solvent used in this study due to its food-grade label. In order to compare the efficiency of the use of 96% ethanol, frequently used solvents, namely chloroform:methanol (2:1, *v*/*v*) and acetone, were also used to evaluate the *C. vulgaris* chlorophyll’s extraction yield (Figure 2). The extraction of chlorophylls *a* and *b* with ethanol allowed a yield of 1.7 mg per one g of dry-weight biomass. The solvent mixture of chloroform:methanol (2:1, *v*/*v*) allowed us to obtain a total chlorophyll content of 7.9 mg/g, agreeing with the literature [19], corresponding to about 5 times more when compared with the extraction with ethanol. The extraction with acetone revealed a lower extraction yield of total chlorophyll content (2.4 mg/g). Although this solvent is used for the extraction of chlorophylls from plants [20], dried *C. vulgaris* did not reveal a particularly interesting efficiency. 

Chl *a* content was higher than Chl *b* content (Figure 2) when extracted with ethanol or chloroform:methanol (2:1, *v*/*v*), which agrees with the higher content of Chl *a* in *C. vulgaris* biomass [21]. This allows using ethanol to obtain a representative extract of the chlorophylls of *C. vulgaris* for further studies. 

### 2.2. C. vulgaris Pigments Identification

To evaluate the pigments composition of the ethanolic extract, a thin layer chromatography (TLC) was performed using two mobile phases with different polarities: petroleum ether:1-propanol:water (100:10:0.25, *v*/*v*/*v*) and *n*-hexane:acetone (7:3, *v*/*v*). The separation was carried out on the principle of affinity of substances to the stationary phase and solubility in the mobile phase according to their polarity as follows: the fastest mobility compound was carotene, pheophytin, Chl *a*, Chl *b*, and xanthophylls (Appendix A), according to the literature [7,22]. The petroleum ether:1-propanol:water eluent allowed the separation and identification of four different pigments (Appendix A), while the less polar eluent, *n*-hexane: acetone, allowed the separation and identification of 5 pigments: the same previously identified with the addition of the pheophytin (Appendix A), identified by its Rf and color [7]. Despite the additional band identification, the separation of the most polar compounds with *n*-hexane: acetone was not as efficient as in the TLC with the organic phase with low water content since a slight overlap of xanthophylls and Chl *b* was observed. Xanthophylls may correspond to lutein and zeaxanthin due to their polarity similitude and due to the broad yellow band verified in the TLC plate on Appendix A, as reported in the literature [7], agreeing with the composition of *C. vulgaris* pigments [23,24].

Albeit different pigments are present, chlorophyll *a,* as the most abundant pigment in *C. vulgaris* and responsible for the dark green color, was used as a diagnostic pigment to study the color stability of *C. vulgaris* pigments in ethanol solutions.

### 2.3. Evaluation of C. vulgaris Color Stability

Since Chl *a*, present in *C. vulgaris* ethanolic extract, is susceptible to chemical degradation, resulting in a decrease in color intensity [25], the evaluation of pigments’ stability under different storage conditions is necessary for further application as a natural colorant in food products. Therefore, this investigation was undertaken to study the influence of temperature (4 and 60 °C), light (in the presence and absence of light), alkaline environment (with or without NaOH), and atmosphere (oxygen-rich or argon-rich atmosphere) on the stability of ethanol solutions of *C. vulgaris* pigments along 9 days (216 h). The evaluation of pigments/green color degradation over time was carried out with 16 experiments in total, measuring the absorbance.

As the maximum absorption of chlorophylls strongly depends on the type of solvent [26], the ultraviolet-visible spectrum (300–700 nm) of *C. vulgaris* ethanolic extract was performed (Appendix A). The wavelength of 418 nm corresponded to the maximum absorbance attributed to the absorption of Chl *a* in ethanol [18], corroborating the majority of this pigment in the ethanolic extract. In this sense, the evaluation of pigments stability over time was recorded at 418 nm.

The decrease of absorbance over time (0, 48, 96, 168, 216 h) for the 16 experimental storage conditions are represented in Figure 3. It is possible to observe that after 9 days of storage, the absorbance decreased for all conditions, showing the instability of the green color. For 4 °C in the dark, the absorbance decrease was between 41 and 54% after 216 h (Figure 3a). With the increase in temperature, the degradation was higher for the same period (73.8% decrease, Figure 3b). In the presence of light, the degradation was even higher (79.4% decrease, Figure 3c), mainly when the temperature was set to 60 °C (89.4% decrease, Figure 3d). The use of a modified atmosphere and alkaline environment seemed not to have a high influence on the degradation of chlorophylls. 

The major absorbance decrease was observed for the first period measured, 48 h of storage. In the conditions that have in common the temperature of 4 °C in the dark, it was observed that with the addition of NaOH in the presence of argon, a lower absorbance decrease (−0.001 Δabs/h) was obtained when compared with the other ones (average of −0.003 Δabs/h) (Figure 3a). Nevertheless, these slopes were lower than those observed for all other conditions, −0.006 Δabs/h at 60 °C in the dark (Figure 3b), −0.007 Δabs/h at 4 °C in the presence of light (Figure 3c), and −0.010 Δabs/h at 60 °C in the presence of light (Figure 3d). Thus, the presence of light and high temperature appears to be the factors that contributed more to the Chl *a* degradation. 

The color of *C. vulgaris* ethanolic extract was also measured using the CIELAB system, which complements the spectrophotometric method since color vision is a complex phenomenon and its measurement can be more complex than absorption at specific wavelengths [12]. Thus, the measurement of −a* is a parameter that has been used to evaluate over time the green color loss of ethanolic solutions [12,18]. Figure 4 represents the decrease of −a* value over time. As verified for the absorbance measurements, it is possible to observe that after 9 days of storage, the −a* decrease for the temperature of 4 °C in the dark was only 51.9% after 216 h (Figure 4a), which was much lower than the observed with 60 °C in the dark, which for the same period presented a decrease of 86.7% (Figure 4b), as well as for the presence of light at 4 °C (Figure 4c), 91.7% decrease, and at 60 °C (Figure 4d), 100% decrease. In agreement with the data obtained for the absorbance measurement, the color decrease was mainly observed for 48 h of storage. 

In order to better understand which conditions have more influence in Chl *a* degradation and to understand the way that those conditions interact between them, an unreplicated 2^4^ full factorial design with two levels was performed for 48 h of storage. The two levels are coded (+1) and (−1) for the higher and lower limits of each one, respectively. The absorbance at 418 nm (Y_1_) and −a* (Y_2_) for the full factorial design is represented in Appendix A.

The Pareto chart (Figure 5a) represents the effects on color stability measured by the absorbance at 418 nm. The linear terms of the variables light (X_2_) and temperature (X_1_) exhibited a significant effect on pigment degradation. This is shown by the bars of the standardized effect that are beyond the vertical red line, representing the statical significance at a 95% confidence level, which does not happen when considering their interactions. Moreover, the Pareto chart evidenced that the linear light variable exerts the most preponderant effect. Accordingly, the Pareto chart in Figure 5b reveals that the linear terms of the variables light (X_2_) and temperature (X_1_) showed a significant effect on the green color vanishing (−a*) of *C. vulgaris* ethanolic extract, being the presence of light the most significant factor as well. According to both Pareto charts (Figure 5), under the studied conditions, the variables atmosphere (X_3_) and the alkaline environment (X_4_) on ethanolic solutions had no statistical impact on the color loss. Contrary to the Pareto chart of the response at Abs 418 nm (Figure 5a), the Pareto chart of the −a* response (Figure 5b) showed significant two 2-way interactions between temperature and light (X_1_X_2_). When an interaction is significant, it means that the effect of a term on the response is distinct at different levels of another independent variable [27]. 

When chlorophyll in organic solvents such as ethanol, acetone, or benzene is exposed to light in the presence of oxygen, it is irreversibly bleached by photo-oxidation, resulting in colorless derivates [28]. The presence of an argon-rich atmosphere, instead of an oxygen-rich atmosphere, did not have a significant effect on the protection of chlorophylls degradation, possibly due to the fact that there are oxygen molecules present in the solution, which could be enough to promote photo-oxidation reactions. The lack of effect of the modified atmosphere was also verified on the stability of pigments (Chl *a*, Chl *b*, and lutein) of pistachio kernels since no differences were observed during storage, irrespective of the use of oxygen scavengers and high gas barrier plastic films [13]. 

Temperature is also a well-established main factor influencing the stability of chlorophylls [11] since high temperatures, such as 60 °C used in this study, could promote the formation of pheophytins, leading to a green color loss [2]. It was also already reported [18] that the rate of color loss of Chl *a* in ethanolic solutions increases with temperature (tested from 20 to 50 °C). Moreover, the stability of Chl *a* or green color with temperature in broccoli and green beans treated from 40 up to 96 °C [29] and spinach puree in a temperature range of 50–120 °C [12] was also affected. The pH also has an influence on the stability of chlorophyll pigments, mainly changing their structures by pheophytinization. When an acidic solution was added to the *C. vulgaris* ethanolic extract, the solution changed immediately from bright green to olive brown (results not shown). However, the addition of NaOH did not significantly influence the chlorophyll degradation in the present study. Alkaline conditions have been reported to induce oxidation of the isocyclic ring and de-esterification of phytol in chlorophylls. These reactions do not significantly affect the color of the product since these compounds retain intact the basic structure of the chromophore group with Mg^2+^ linked to the porphyrin ring [30]. Consequently, chlorophyll is reported to be stable at alkaline conditions [12] or even more stable, as verified for the chlorophylls in coriander leaf puree that was found to be most heat stable at pH 7.5 [16].

To evaluate the effect of pigment concentration in the *C. vulgaris* ethanolic extract in different storage conditions, another set of experiments (set 2) was performed, starting with a higher-concentrated ethanolic pigment solution. Since the presence of NaOH seemed to prevent the degradation of long-term chlorophyll at 4 °C in the dark under air atmosphere (Figure 3a), the argon-rich atmosphere was no longer tested. Moreover, 28 °C was tested instead of 60 °C since higher temperatures greatly promoted Chl *a* degradation. The first batch of experiments also revealed that after 48 h of storage, most experiments already had a faint green color; thus, in the second set, beyond the higher concentration of pigments in the ethanol *C. vulgaris* extract, the time span of the experiments was enlarged with points at 9.5, 36, 65.5, 138, 155.5, 184, 228.5, and 324 h. Thereafter, a total of eight storage conditions were tested throughout the measurement of the absorbance at 418 nm and −a* value (Figure 6). According to Figure 6a,b, the lower decrease of Abs at 418 nm occurred in the absence of light, with the two temperatures (4 °C and 28 °C) seeming to have no impact on the pigment’s degradation. Moreover, the addition of NaOH also appeared to have no influence on the maximum absorbance at 418 nm, as the slower decrease of absorbance with time observed in the initial 100 h in the presence of light was no longer present for longer storage periods. The measurement of −a* (Figure 6c,d) also revealed the major importance of the dark to protect chlorophyll degradation, with the addition of NaOH showing a positive effect, although in less extent, for long periods of time (Figure 6c). This protective effect is also noticed in the presence of light (Figure 6d).

To understand the influence of pigments concentration on their degradation rate, the degradation at 4 °C with and without NaOH in the dark and in the presence of light was evaluated. Hence, the analyzed parameters at 48 h were estimated (Table 1).

The samples with higher chlorophyll concentration (set 2) had a higher degradation rate of Chl *a* (A418) and green color (−a*), particularly in the presence of light, with higher slopes observed (Table 1). In dark conditions, the slopes were much lower and similar between sets, allowing us to infer the stabilization of chlorophylls in these ethanolic solutions.

The full factorial analysis was employed at 9.5 h and at 65.5 h, and the responses Y_1_ (abs 418 nm) and Y_2_ (−a*) are represented in Appendix A. Full factorial design allowed us to demonstrate the statistical relevance (at a 95% confidence level) of the presence of light (X_2_) on the decreasing absorbance (418 nm) at 9.5 h of storage, as shown in the Pareto chart (Figure 7a), and the statistical irrelevance of the other linear terms, namely temperature (X_1_) and alkaline environment (with or without NaOH, X_3_). For the same time of storage (9.5 h) of *C. vulgaris* ethanolic extract, similar behavior was observed in the stability of the −a* parameter (Figure 7b), where only the presence of light had a slight influence. After a longer period of storage (65.5 h), the presence of light greatly influenced the degradation of pigments and green color in the ethanolic solution, verified by the statistical relevance on both absorbance and −a* parameter, as demonstrated by the Pareto charts in Figure 7c,d. Moreover, after this period, the linear variable temperature (X_1_) also exhibited a lower preponderant influence on the decrease of absorbance without statistically affecting the decrease of −a*. The evaluation of the CIELAB parameter −a* reflects the vanishing of green color, which is a crucial visual parameter to take into account for further application of these colorants in the food industry. 

Significant two 2-way interactions between the terms light (X_2_) and temperature (X_1_) were not observed, contrarily to the experiments performed at 60 °C, allowing us to infer that the temperature of 28 °C did not promote a significant extent of chlorophyll degradation as observed at 60 °C. It was reported that both refrigerator and room temperatures are suitable to store Chl *a* dissolved in acetone, which was kept for 84 days in the dark [17]. Moreover, the degradation rate of Chl *a* of pistachio kernels was only slightly different at 10 and at 25 °C [13].

### 2.4. Degradation Kinetic of Green Color at Different Temperatures

The CIELAB −a*/−a_0_* value is a good way to measure chlorophyll degradation and color loss with temperature [31]. Using linear regression, the data were analyzed to determine the overall order and rate constant for the degradation reaction of green pigments. Accordingly, ln (−a*/−a_0_*) was plotted *vs.* time (0, 6, 24, 29, 47, and 102 h), from which the rate constants (*k*) were calculated. Figure 8a shows the representative values for the first-order plots for the vanishing of greenness (−a*) for *C. vulgaris* ethanolic extract preserved in the dark at different temperatures, namely 4, 15, 28, 45, and 60 °C. The correlation coefficient was >0.84 in all cases, confirming that the fading of the visual green color followed first-order kinetics at all temperatures. These results are in line with the first-order reaction found for Chl *a* degradation in solutions with different percentages of ethanol (1–60%) [18] and in vegetables, such as spinach puree [12], broccoli juice [32], and green peas [9].

From the rate constants obtained in Figure 8a, it was possible to develop a semi-logarithmic plot of *k vs.* the inverse of temperature. The Arrhenius plot for color loss of *C. vulgaris* pigments in ethanol revealed that the main color loss occurred between 28 and 60 °C under the studied conditions, emphasizing the importance of the higher temperatures in the green color degradation. When plotting the three higher temperatures, it was obtained an activation energy of 74 kJ/mol (Figure 8b). As no significant degradation has been observed for the temperatures of 4 °C and 15 °C, these values were not taken into consideration for the calculation of the activation energy of chlorophyll degradation. This emphasizes the stability of pigments below room temperature and their higher degradation rate at temperatures higher than room temperature in ethanol solutions.

The activation energy of chlorophyll degradation obtained is of similar magnitude to the ones determined in previous studies, where it was reported an activation energy from 18.4 to 85.1 kJ/mol for different ethanol concentrations [18]. However, most of the studies of chlorophyll degradation have been done in whole foods, not in isolated chlorophylls. Activation energies of 28.7 kJ/mol have been reported for spinach puree, 41.6 kJ/mol for mustard leaves, and 34.0 kJ/mol for mixed puree (75 and 115 °C) [33]. Activation energies for color loss in broccoli juice for temperatures between 80 and 120 °C were reported as 71.0 kJ/mol [32], 11.4–16.0 kJ/mol for green chili puree (60 and 90 °C) [34], and 34.0–49.8 kJ/mol for blanching of green peas (70 and 100 °C) [9]. These low values at high temperatures show that the degradation of chlorophylls when in the native matrix is minimized, although variations for the different matrices may occur. Beyond the influence of the matrices, the temperature ranges used in these studies have also influenced the determination of the activation energies for chlorophylls’ degradation.

### 2.5. Food Application

As a request from the food industry to have green-colored rice used in sushi dishes, the green *C. vulgaris* ethanolic extract (1 and 2 mL) was added to cooked cold white rice. The visual color stability was observed over time at 4 °C and is shown in Figure 9. At time 0 (Figure 9a), the addition of chlorophyll extract, principally the addition of 2 mL, originated an appealing bright green rice. Only after 3 days of storage (Figure 9b), the rice started to lose its bright green color. The color changed to a slight olive green at 7 and 13 days of storage, possibly due to the formation of pheophytins since the pH was not controlled [2]. The color of the rice in the presence of light remained visually unchanged (Figure 9), similar to the rice maintained in the dark. Thus, although light has a great influence on the fading of color of *C. vulgaris* ethanolic extract, it does not have an influence when applied to the rice. The application of the green extract in cooked cold rice proved the potential of *C. vulgaris* as a source of a food green color additive, mainly for the immediate addition and consumption of processed food, meeting the need of consumers for natural colorants. 

## 3. Materials and Methods

### 3.1. Extraction

Different solvents and extraction methodologies were tested to obtain the pigments (green color) from *C. vulgaris* microalgae dry biomass. Thus, 10 mL of ethanol 96%, acetone, or chloroform:methanol (2:1, *v*/*v*) were added to 100 mg of biomass, and the respective suspensions were homogenized for 10 min at room temperature. The suspensions of all extractions were centrifuged for 3 min (4000× *g* rpm), and the supernatant was collected. Two other extractions with 5 mL were performed, and all supernatants were combined.

### 3.2. Total Chlorophylls Content Determination

Absorption measurements of chlorophyll were made on an Eon Microplate Spectrophotometer (BioTek Instruments, Inc. Winuski, VT, USA). The equations proposed by Wellburn [26] and Lichtenthaler [35] were used for the determination of chlorophyll concentration in the extracts of *C. vulgaris* at one of the maximum absorption wavelengths that slightly changed according to the organic solvent used. The absorbance was measured at 649 and 664 nm to determine chlorophyll *a* (Chl *a*) and *b* (Chl *b*) contents when ethanol was used [35]. For acetone extract, the absorbance was measured at 645 and 662 nm [35]. For chloroform:methanol (2:1, *v*/*v*) extract, the absorbance was measured at 648 and 666 nm [26]. The total chlorophyll content was determined as the sum of Chl *a* and Chl *b* and expressed as mg/g biomass dry weight [1].

### 3.3. Pigments Identification by Thin Layer Chromatography (TLC)

To identify the pigments of *C. vulgaris* extracted with ethanol 96%, a thin layer chromatography (TLC) was performed. Thus, *C. vulgaris* ethanolic extract was applied to 4 cm × 10 cm silica plates. Two different eluents were used, namely petroleum ether:1-propanol:water (100:10:0.25, *v*/*v*/*v*) and *n*-hexane:acetone (7:3, *v*/*v*). 

### 3.4. Evaluation of Pigments Stabilization and Color Loss

*C. vulgaris* pigments were extracted with ethanol to evaluate their storage stability under different conditions, namely temperature, light, atmosphere, and alkaline environment. In the data set, 1.4 g of *C. vulgaris* biomass was mixed with 300 mL of ethanol 96% and stirred for 15 min to extract pigments. Thereafter, the ethanolic solution was centrifuged and filtered with glass fiber filters under a vacuum. More ethanol 96% was added until the absorbance at 418 nm reached 0.6 (addition of about 800 mL ethanol). NaOH 1 M (100 μL) was added to 300 mL of the solution. Since the solutions were not aqueous, the pH value that could be measured by a pH meter would not be reliable. For that reason, the variation in the concentration of NaOH in the ethanolic solutions was taken as a measure of the concentration of variation of H^+^ in the samples.

The ethanolic extracts were distributed in glass bottles (25 mL). Eight bottles (four with and four without NaOH addition) were placed at 4 °C (refrigerator temperature) and the other eight bottles at 60 °C (a representative extremely high temperature) in a bath with paraffin (to avoid long-term evaporation). At each temperature, four bottles were protected from the light (two with and two without NaOH addition), and four bottles were subjected to light with a photoperiod of 24 h (about 3500 lx). In each temperature, half of the bottles were under an air atmosphere, and the other half were under an argon atmosphere, according to the scheme represented in Appendix A. The absorbance (300–700 nm, Jenway 6405 UV/Vis, Cole-Parmer Ltd., Vernon Hills, IL, USA) and color through CIELAB system (PerkinElmer, Lambda35-UVWinLab program to obtain the transmittance spectra at 380–780 nm, and COLOR-UVWinLab program for CIE L*a*b*determination, illuminant D65, 10°) of each solution were measured along 9 days (at 0, 48, 96, 168, and 216 h). CIELAB system expresses color in a three-dimensional space, with three axes: L* for the lightness from black to white, a* from green (−) to red (+), and b* from blue (−) to yellow (+).

To statistically analyze the data, an unreplicated 2^4^ full factorial design with two levels was used to evaluate, after 48 h of storage, the effect of temperature (X_1_, 4 °C, and 60 °C), light (X_2_, presence or absence of light), modified atmosphere (X_3_, presence or absence of oxygen), and alkaline environment (X_4_, with or without NaOH). The two levels are coded (+1) and (−1) for the higher and lower limits of each one, respectively. In a two-level full factorial design, 2^k^ runs are required, where k represents the number of factors to be analyzed, which results in 16 runs performed. The experimental data were statistically analyzed using Minitab v17 software. The Pareto charts were developed for the linear terms and for their interactions that showed statistical relevance to simplify the model. The Pareto chart with the linear and all the 2-way interaction terms are presented in Appendix A. 

The second set of experiments (data set 2) was performed with an unreplicated 2^3^ full factorial design with two levels to evaluate the effect of temperature (X_1_, 4 °C and 28 °C), light (X_2_, presence or absence of light), and alkaline environment (X_3_, with or without NaOH) for 9.5 and 65.5 h, using a more concentrated *C. vulgaris* ethanolic solution (8 g of *C. vulgaris* biomass extracted with 600 mL of 96% of ethanol), resulting in 8 runs performed. The preparation of the bottles with the ethanolic extract was performed as described for data set 1, except for the use of a water bath to monitor the temperature at 28 °C instead of a bath with paraffin. The absorbance and color stability were evaluated over 14 days (at 0, 9.5, 19, 36, 65.5, 138, 156, 184, 229, and 324 h, Appendix A).

### 3.5. Degradation Kinetic of Green Color

The bottles with ethanolic extract (data set 2) were placed at five different temperatures: 4, 15, 28, 45, and 60 °C (Appendix A), protected from the light. The color of each solution was measured over 4 days (0, 6, 24, 29, 47, and 102 h). Since green is the major color of *C. vulgaris* ethanolic extract, the increase of parameter “a*” values from a more negative value towards zero (−a*) was considered a visual parameter to describe the green color degradation at different temperatures [12]. The first-order reaction rate constant (*k*) was calculated using the following equation:(1)$$\ln\left(\frac{-a*}{-a{*}_{0}}\right)=-kt$$
where −a* is the “−a*” value measured at different times (*t*) and $-a{*}_{0}$ is the “−a*” value measured at time zero. 

Temperature dependence of green color degradation was determined by the Arrhenius equation:(2)$$k=k_{0}e^{-E_{a}/RT}$$
where *E_a_* is the activation energy (kJ mol^−1^), *k* is the first-order reaction rate constant (s^−1^), *k*_0_ is the pre-exponential factor, R is the universal gas constant (8.3145 J mol^−1^ K^−1^), and *T* is the absolute temperature (*K*).

### 3.6. Food Application of C. vulgaris Pigments

An ethanolic extract of *C. vulgaris* was obtained using 1 g of biomass and 10 mL of ethanol 96%. This extract (1 mL) was added to two containers with 18 g of cold-cooked rice, the minimum volume necessary to confer an appropriate coloration to the rice. Moreover, 2 mL was also added to two containers with the same amount of rice to evaluate the effect of the chlorophyll concentration in the applied food product. A fifth container was used as a control, only containing the cooked rice. All containers were placed at 4 °C, two in the presence of light in a 24 h photoperiod and two in the absence of light; the control was maintained at 4 °C in the presence of light (Appendix A). The color was visually evaluated through photographs taken along 13 days.

## 4. Conclusions

This work allowed us to obtain a food-grade and stable extract to be applied as a green colorant ingredient, pursuing the market tendency for clean-label products. The assessment of the color stability of a *C. vulgaris* ethanolic extract, composed mainly of Chl *a* in different storage conditions, allowed us to conclude that light was the main variable that negatively affected the green color, followed by temperature, under the studied conditions. The alkaline environment and an inert atmosphere (argon-rich atmosphere) had no statistical effect on the green color preservation. The loss of color in the ethanol solution with temperature followed the first-order kinetic, being more significant between 28 and 60 °C, with an activation energy of 74 kJ/mol. These results showed that *C. vulgaris* chlorophylls could be preserved in a food-grade solvent at room or lower temperatures when kept in the dark, obtaining an easy-to-handle extract that can be used as a natural food ingredient without conferring an unpleasant flavor. Moreover, the green residue left is also a non-odorant food ingredient rich in protein, starch, and cell wall polysaccharides, thus with high potential to be further valued. 

## Figures and Tables

**Figure 1 molecules-28-00408-f001:**
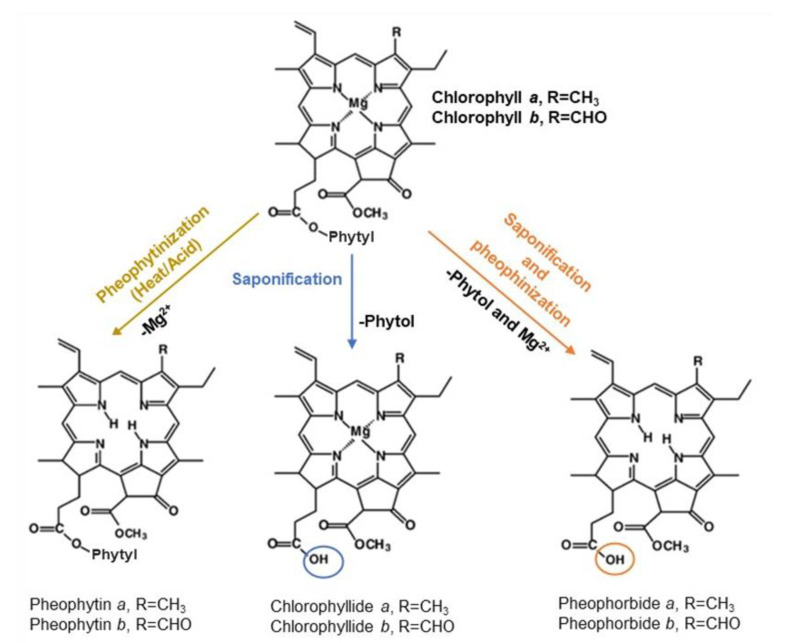
Major Chl *a* and Chl *b* degradation routes through pheophytinization and saponification.

**Figure 2 molecules-28-00408-f002:**
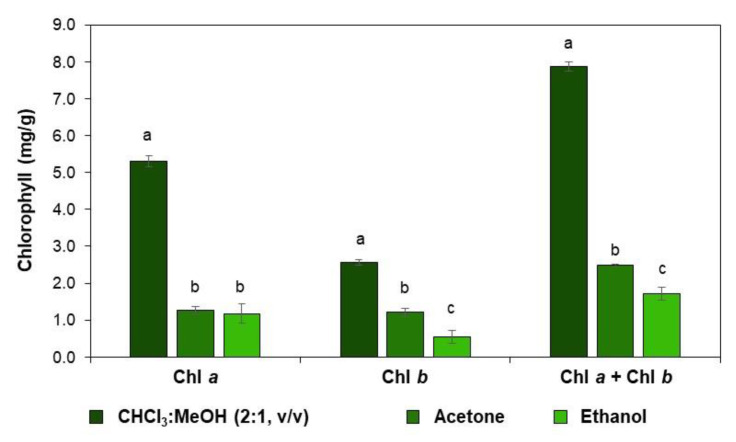
Extraction yield of chlorophyll *a* (Chl *a*, mg/g), chlorophyll *b* (Chl *b*, mg/g), and total chlorophylls (Chl *a* + Chl *b*, mg/g) from *C. vulgaris* biomass, using chloroform:methanol (2:1, *v*/*v*), acetone, and ethanol (96%). Mean values ± SD, n = 9. The different characters above the bar indicate statistical differences (*p* < 0.05) between compared groups (One-Way ANOVA, Tukey’s multiple comparisons test).

**Figure 3 molecules-28-00408-f003:**
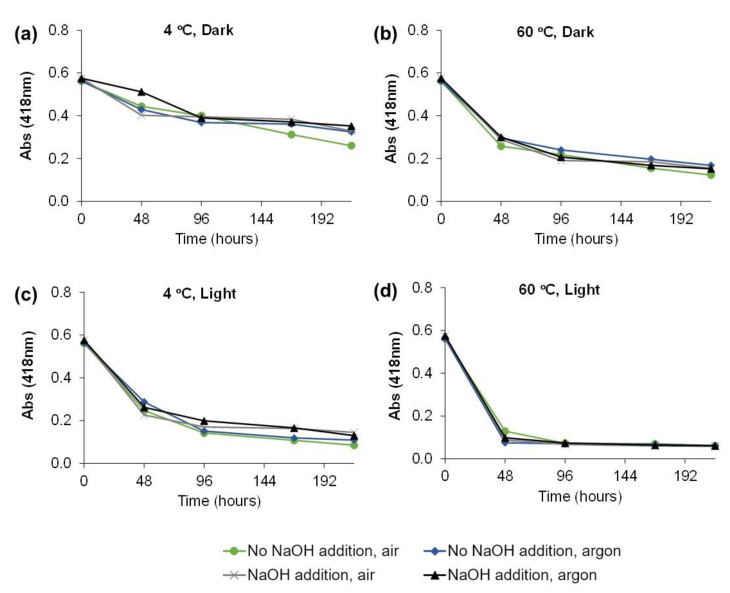
Absorbance (418 nm) measured over time for the 16 experimental conditions (**a**) 4 °C, Dark, (**b**) 60 °C, Dark, (**c**) 4 °C, Light, (**d**) 60 °C, Light.

**Figure 4 molecules-28-00408-f004:**
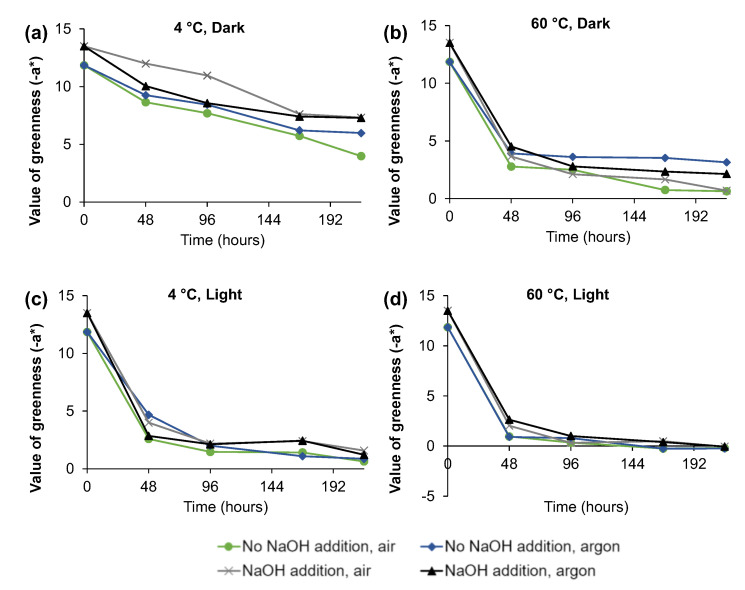
Value of greenness (−a*) measured over time for the 16 experimental conditions (**a**) 4 °C, Dark, (**b**) 60 °C, Dark, (**c**) 4 °C, Light, (**d**) 60 °C, Light.

**Figure 5 molecules-28-00408-f005:**
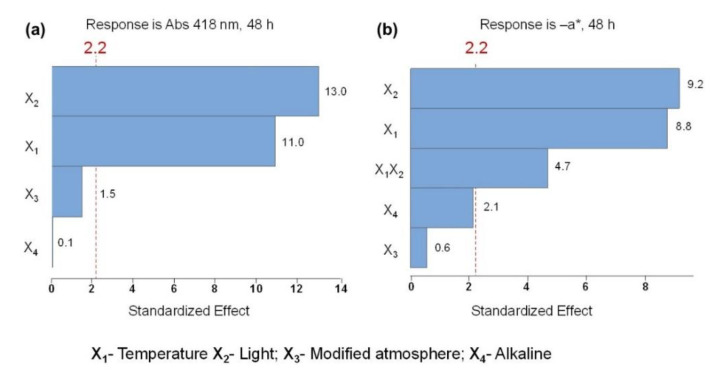
Pareto charts of the standardized effects: (**a**) response is Abs (418 nm), 48 h (*p* < 0.05); (**b**) response is the value of greenness (−a*), 48 h (*p* < 0.05).

**Figure 6 molecules-28-00408-f006:**
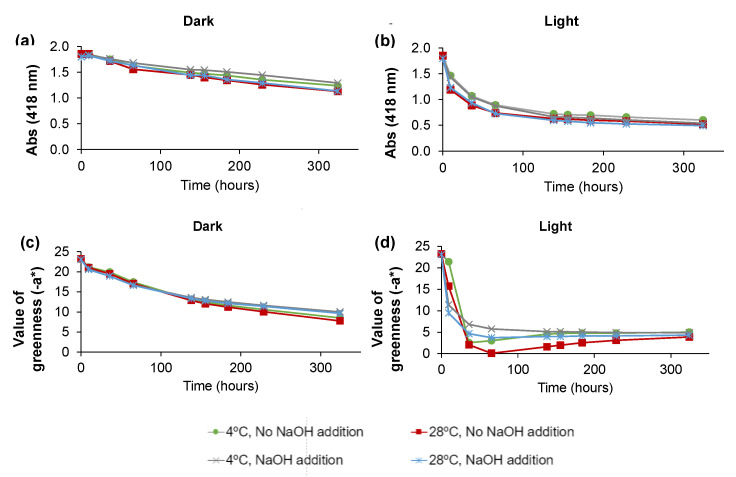
Absorbance (418 nm) measured over time for the 8 experimental conditions (**a**) in the dark, and (**b**) in the light. Value of greenness (−a*) measured over time for the 8 experimental conditions (**c**) in the dark, and (**d**) in the light.

**Figure 7 molecules-28-00408-f007:**
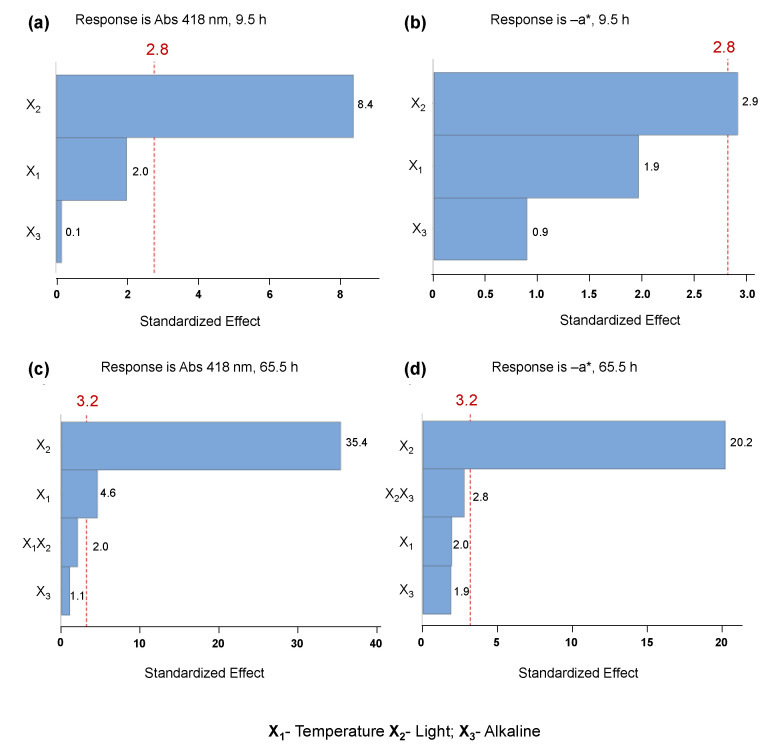
Pareto charts of the standardized effects: (**a**) response is Abs (418 nm), 9.5 h (*p* < 0.05); (**b**) response is the value of greenness (−a*), 9.5 h (*p* < 0.05). Pareto charts of the standardized effects: (**c**) response is Abs (418 nm), 65.5 h (*p* < 0.05); (**d**) response is −a*, 65.5 h (*p* < 0.05).

**Figure 8 molecules-28-00408-f008:**
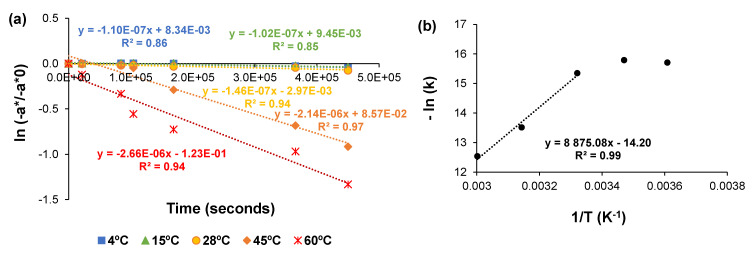
(**a**) First order plot of color (−a*) degradation in *C. vulgaris* ethanolic solution at different temperatures; (**b**) Arrhenius plot for color loss of ethanolic solutions of *C. vulgaris* chlorophylls.

**Figure 9 molecules-28-00408-f009:**
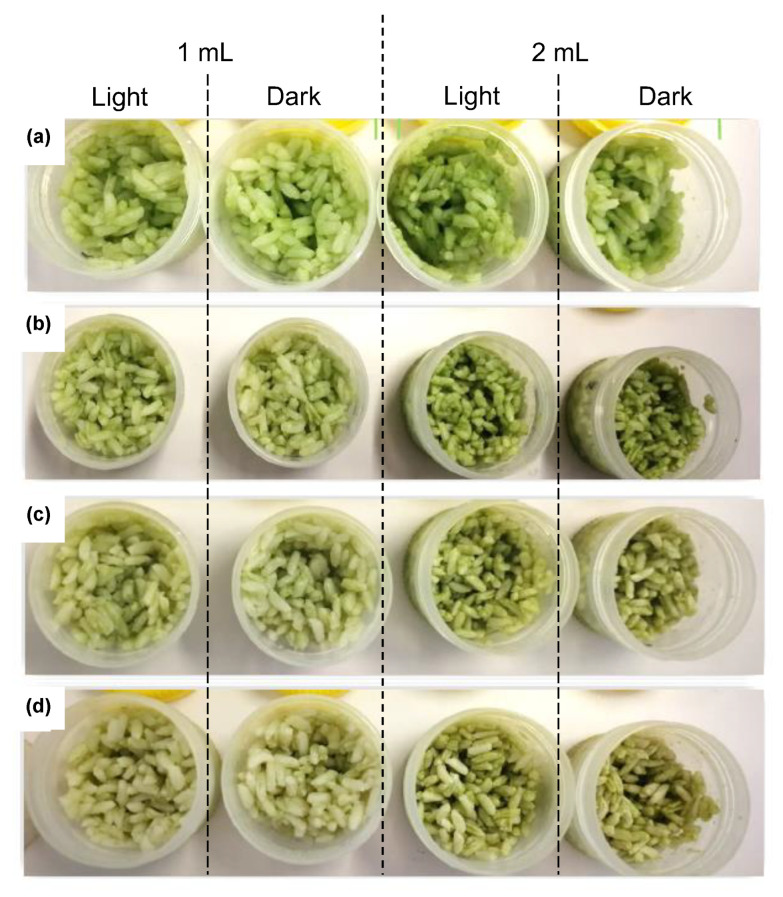
Addition of 1 and 2 mL of *C. vulgaris* ethanolic extract to white rice and evaluation of visual color during storage at 4 °C in the presence and absence of light at (**a**) day 0; (**b**) day 3; and (**c**) day 7, and (**d**) day 13.

**Table 1 molecules-28-00408-t001:** Degradation rate (slope) of Chl *a* and green color (−a*) of 4 experiments of each set of experiments at 4 °C for the first 48 h. Set 1 refers to the first batch of experiments (lower concentration, 0.6 of initial absorbance), set 2 refers to the second batch of experiments (high concentration, 1.8 of initial absorbance), and %dif refers to the difference between both sets expressed in percentage.

Conditions	Runs	A418			−a*		
		Set 1	Set 2	%dif	Set 1	Set 2	%dif
No NaOH addition, dark	1	−0.0025	−0.0029	13.8	−0.0667	−0.0892	25.2
No NaOH addition, light	2	−0.0066	−0.0163	59.5	−0.1931	−0.4264	54.7
NaOH addition, dark	3	−0.0036	−0.0012	−66.7	−0.0313	−0.1048	70.1
NaOH addition, light	4	−0.0073	−0.0170	57.1	−0.1981	−0.3470	42.9

## Data Availability

All data are available in the manuscript.

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
