# Peer review of "Stabilization of Natural Pigments in Ethanolic Solutions for Food Applications: The Case Study of Chlorella vulgaris"

_molecules, 2023, doi:10.3390/molecules28010408_

Round 1

Reviewer 1 Report

The paper is very interesting and well-written. The language is very good. In general, the paper is dedicated to the stability of chlorophyll. This issue is topical considering the instability of chlorophyll and accordingly the loss of the green color.

However, it is  necessary to improve some statements

1. Specify the units of yields in section 2.1 - mg of chlorophyll per 1 g of biomass? In  lines 45-46 it is going about % of the dry weight

2.  lines 133-136 there is a good explanation about the usage of 418 nm. The maximum. OK. However, in section 3.2 we have 649 and 664 nm.  Specify. Moreover, I propose to make  a small statement that maximums do not depend on the solvent in section 3.2 (for ethanol, acetone, and chloroform).

3. The work is saturated by good descriptive information about the stability of chlorophyll. It is strongly recommended to add the reaction of the degradation of chlorophyll under the influence of different factors - light, temperature, etc.

4. It is recommended to add information in the Conclusion section about the influence of different factors on the stability of chlorophyll 

Author Response

We thank the Reviewer valuable comment regarding the improvement of our manuscript. The answer to the reviewer concern is described in detail above and the manuscript was revised accordingly. We hope that the revised form of the manuscript is suitable for publication.

"The paper is very interesting and well-written. The language is very good. In general, the paper is dedicated to the stability of chlorophyll. This issue is topical considering the instability of chlorophyll and accordingly the loss of the green color.

However, it is necessary to improve some statements"

1. Specify the units of yields in section 2.1 - mg of chlorophyll per 1 g of biomass? In lines 45-46 it is going about % of the dry weight

Yes, in section 2.1 the yields are expressed in mg of chlorophyll per 1 g of biomass. The units were now specified in the manuscript. In the introduction, the units were also converted to mg of chlorophyll per one g of biomass.

2. Lines 133-136 there is a good explanation about the usage of 418 nm. The maximum. OK. However, in section 3.2 we have 649 and 664 nm. Specify. Moreover, I propose to make a small statement that maximums do not depend on the solvent in section 3.2 (for ethanol, acetone, and chloroform).

Chlorophyll a has two absorption peaks: one at 418 nm and the other at 664 nm. The work was based on the absorbance at 418 nm since it is the maximum absorbance peak. However, the statistical analysis was also performed for 664 nm (data not showed), obtaining the same conclusions. The yield of extraction was determined based on literature that correlates the absorption at 649 nm, which corresponds to one of the wavelength maximum of Chl b, and 664 nm, which corresponds to one of the wavelength maximum of Chl a,with the concentration of chlorophylls a and b. According to literature, the wavelength maxima slightly change with different organic solvents. This information was added to manuscript in the section 3.2.

3. The work is saturated by good descriptive information about the stability of chlorophyll. It is strongly recommended to add the reaction of the degradation of chlorophyll under the influence of different factors - light, temperature, etc.

More information about the degradation reactions of chlorophylls was added to the Introduction section of the manuscript.

4. It is recommended to add information in the Conclusion section about the influence of different factors on the stability of chlorophyll 

The requested information was added to conclusion section.

Reviewer 2 Report

I had the opportunity to review the Submission Molecules-2057901. The paper is about evaluate the green chlorophyll solutions as food ingredients. This study aim to set the most appropriate storage conditions of C. vulgaris ethanolic extract to stabilize the color to further allow their application as a suitable green coloring ingredient without affecting the organoleptic characteristics of the food products. The study is of interest but needs to be completed. I consider that it presents limitations, so my recommendation is major revision.

Main aspects that support my recommendation

Abstract:

i) Line32: C. vulgaris ethanolic extract showed to be a natural food additive to color foodstuffs. Nonetheless, the study aim to set the most appropriate storage conditions of C. vulgaris ethanolic extract to stabilize the color. This statement is not relevant to the purpose of the study.

Introduction:

i) Line60-64: This part is not detailed enough.

ii) Line65: not yet evaluated? pH control, enzymatic treatments, temperature control, addition metal ions. This efforts in preserving the green chlorophylls mostly in foods have been carried. not yet evaluated?

Results and Discussion:

i) Line80-81: extracted with CHCl3:MeOH (2:1, v/v), total Chlorophyll about 8.0 mg/g; however, extracted with ethanol, total Chlorophyll about 1.5 mg/g. The difference between the two values is about 5 times. Ethanol is suitable as extraction reagent?

ii) Figure 1: A variance analysis should be performed for all groups.

iii) Line122: Evaluation of C. vulgaris color stability. I think thin layer chromatography experiment should add a group of extracts: extracted with CHCl3:MeOH (2:1, v/v).

iv) Figure 2, No duplicate samples? Abs? not OD value? No significance analysis?

v) Figure 3, ‘Value of greenness’ replace ‘-a*’. No unit of this value?

vi) Line 214-215, already reported temperature (20-50℃), why does the author use 4 and 60℃? And only two temperature? is it reasonable?

vii) Figure 5, No duplicate samples? No significance analysis? The picture is not clear, low resolution.

ix) Figure 7, the font format in the figure is inconsistent. Figure 7b, I think this fitting result is too far fetched.

x) Line350 & Figure 8, why set up this experimental group (1 and 2 mL)? no sense.

Materials and Methods:

i) Line 401-402, what is the pH value of the solution? Why only set up a group of experiments?

ii) Line 403-404, why does the author use two temperature?

iii) Light and Dark, what is the light intensity? Why not set different light intensity?

References:

Inconsistent format of references:

Line 510, Line 517, Line 526, Line 535, Line 541, Line 556, Line 569, Line 575 ([34]).

Author Response

We thank the Reviewer valuable comment regarding the improvement of our manuscript. The answer to the reviewer concern is described in detail above and the manuscript was revised accordingly. We hope that the revised form of the manuscript is suitable for publication.

"I had the opportunity to review the Submission Molecules-2057901. The paper is about evaluate the green chlorophyll solutions as food ingredients. This study aim to set the most appropriate storage conditions of C. vulgaris ethanolic extract to stabilize the color to further allow their application as a suitable green coloring ingredient without affecting the organoleptic characteristics of the food products. The study is of interest but needs to be completed. I consider that it presents limitations, so my recommendation is major revision.

Main aspects that support my recommendation"

Abstract:

  1. i) Line32: vulgaris ethanolic extract showed to be a natural food additive to color foodstuffs. Nonetheless, the study aim to set the most appropriate storage conditions of C. vulgaris ethanolic extract to stabilize the color. This statement is not relevant to the purpose of the study.

The last part of the abstract was rephrased: “Therefore, this work showed that C. vulgaris chlorophylls can be preserved in ethanolic solutions at room or lower temperatures when protected from light, allowing to obtain a suitable natural food ingredient to color foodstuffs.”

Introduction:

  1. i) Line60-64: This part is not detailed enough.

More information, namely regarding to the degradation routes of chlorophylls was added to the manuscript in the introduction section.

  1. ii) Line65: not yet evaluated? pH control, enzymatic treatments, temperature control, addition metal ions. This efforts in preserving the green chlorophylls mostly in foods have been carried. not yet evaluated?

Indeed, this phrase is not well explicit. It meant that the studies that evaluate the green color stability on solutions did not further evaluate their potential as food colorant ingredient. This part was removed from the manuscript.

Results and Discussion:

  1. i) Line80-81: extracted with CHCl3:MeOH (2:1, v/v), total Chlorophyll about 8.0 mg/g; however, extracted with ethanol, total Chlorophyll about 1.5 mg/g. The difference between the two values is about 5 times. Ethanol is suitable as extraction reagent?

We agree with the referee considering the poor extractability of ethanol when compared to CHCl3:MeOH (2:1, v/v). However, as stated, this was the solvent elected in this study due to its food grade label.

  1. ii) Figure 1: A variance analysis should be performed for all groups.

The variance analysis was performed for all groups.

iii) Line122: Evaluation of C. vulgaris color stability. I think thin layer chromatography experiment should add a group of extracts: extracted with CHCl3:MeOH (2:1, v/v).

The thin layer chromatography was performed with the purpose of the evaluation of the ethanolic extract composition to be applied to foods. Since the extraction with CHCl3:MeOH (2:1, v/v) was only performed to understand how much less chlorophylls were extracted with ethanol, no further characterization of this extract have been performed because it was not a food grade ingredient.

  1. iv) Figure 2, No duplicate samples? Abs? not OD value? No significance analysis?

Indeed, these analyses were not performed in duplicate, since sixteen different conditions were already evaluated. However, the data obtained in some conditions validate the data obtained in other conditions. For example, if it was not possible to observe differences using modified atmosphere with NaOH addition, it was not expected to observe differences with no NaOH addition. Absorbance at the maximum point and the value of greenness showed to be a proper measurement to follow the impact of the different storage conditions on the green color.

  1. v) Figure 3, ‘Value of greenness’ replace ‘-a*’. No unit of this value?

Value of greenness replaced “-a*” in Figure 3 and Figure 5 of the manuscript. This value is dimensionless (no units).

  1. vi) Line 214-215, already reported temperature (20-50℃), why does the author use 4 and 60℃? And only two temperature? is it reasonable?

The temperature 4 ºC was settled since it is the temperature of the fridges, and, consequently, it is easy to keep the solutions at this temperature. Thus, it was stablished a lower temperature (4 ºC) and then an extreme higher temperature compared with what is already observed in literature for stabilization of chlorophylls in ethanolic solutions. In the second set of experiments other temperatures, namely 4 ºC and 28 ºC were also evaluated. Moreover, the degradation kinetics with temperature were also estimated for 4, 15, 28, 45 and 60 ºC. In section 3.4 this information was added.

vii) Figure 5, No duplicate samples? No significance analysis? The picture is not clear, low resolution.

As explained in question iv, these analyses were not performed in duplicate, since a large number of different conditions have been evaluated and some validated the other ones. The picture was replaced to increase resolution.

  1. ix) Figure 7, the font format in the figure is inconsistent. Figure 7b, I think this fitting result is too far fetched.

The font format of Figure 7 was changed, and it is now consistent. In Figure 7b only the temperatures 60, 45, and 28ºC were used to the fitting since it was estimated the activation energy for the degradation of chlorophylls. As at 4ºC and 15ºC the degradation is almost 0, the fitting with these two temperatures has not been performed. This was explained in the text.

  1. x) Line350 & Figure 8, why set up this experimental group (1 and 2 mL)? no sense.

Previous results (data not shown) showed that, for the concentration of the extract used, it would be necessary to add at least 1 mL of the extract for a proper rice coloration. Moreover, the double of the extract was also added to evaluate the effect of the chlorophyll’s concentration in the applied food product. This information was added to manuscript in section 3.6.

Materials and Methods:

  1. i) Line 401-402, what is the pH value of the solution? Why only set up a group of experiments?

The pH was not possible to measure by a pH meter since it was not an aqueous solution. So, for that reason the variation in the concentration of NaOH was explained by the terms “with or without NaOH addition”. This information was added to the manuscript.

Another group of samples was also performed with the addition of acid, but the solutions turned immediately green olive due to the pheophytization of chlorophylls. Consequently, this group of experiments was not analyzed due to the lack of a bright green color at time 0, needed for the further food applications.

  1. ii) Line 403-404, why does the author use two temperature?

As explained in question vi, the temperature 4 ºC was settled since it is the temperature of the fridges and 60 ºC was settled as extreme higher temperature.

iii) Light and Dark, what is the light intensity? Why not set different light intensity?

The light intensity was about 3500 lx in the present of light and 0 lx in the dark. As for the temperatures, these conditions were settled to have the two extremes.

References:

Inconsistent format of references:

Line 510, Line 517, Line 526, Line 535, Line 541, Line 556, Line 569, Line 575 ([34]).

The modifications were made in the manuscript.

Round 2

Reviewer 2 Report

Thank you for submitting the revised version of your paper.Having assessed the responses and the changes made the paper has indeed been improved, and there is the greater clarity that was necessary.

Author Response

We thank the reviewer comment.